# ORF3 Gene of Porcine Epidemic Diarrhea Virus Causes Nuclear and Morphological Distortions with Associated Cell Death

**DOI:** 10.3390/v17111468

**Published:** 2025-11-01

**Authors:** Ndirangu A. Kamau, Jae-Rang Rho, Eui-Soon Park, Jung-Eun Yu, Ji-Yun Yu, Gianmarco Ferrara, Hyun-Jin Shin

**Affiliations:** 1Laboratory of Infectious Diseases, College of Veterinary Medicine, Chungnam National University, Daejeon 34134, Republic of Korea; antonykamau@tukenya.ac.ke; 2Department of Biological Sciences, Technical University of Kenya, Haile Selassie Avenue, Nairobi P.O. Box 52428-00200, Kenya; 3Department of Microbiology & Molecular Biology, College of Bioscience & Biotechnology, Chungnam National University, 220 Gungdong, Yuseong, Daejeon 34134, Republic of Korea; jrrho@cnu.ac.kr (J.-R.R.); khanes@naver.com (E.-S.P.); love_ddalki@hanmail.net (J.-E.Y.); jiyunyoo3@hanmail.net (J.-Y.Y.); 4Department of Veterinary Science, University of Messina, Polo Universitario dell’Annunziata, 98168 Messina, Italy

**Keywords:** PEDV, ORF3, truncated, transmembrane, nuclear

## Abstract

There is increasing research interest in the ORF3 accessory protein of PEDV as a critical element for viral virulence. Here, wild type ORF3 (ORF3_wt_) gene was constructed in pEGFP-C1 vector. Additionally, two truncation mutants, ORF3-N (1-98 amino acids [aa]) and ORF3-C (99-224 aa) were inserted in the same vector. Results of ORF3 expression revealed early cytoplasmic localization but 12 h after transfection, ORF3 accumulated around the nucleus, especially ORF3-N. This caused chromosome condensation and morphological distortion that culminated in cell death. In comparison with the native cells expressing GFP alone, ORF3wt-induced lethality was 6.61% above baseline while ORF3- C expression resulted in moderate increase in cell death (0.64%). ORF3-N was affected the most with 220.32% increased lethality. It was, therefore, inferred that the ORF3 gene encodes a protein that causes nuclear damage, distorts cell morphology and leads to cell death. Furthermore, the role of the protein could be inherent in the N-terminal domain, which consists of the transmembrane domains. These findings underpin the importance of ORF3 gene expression in the host and are rudimental insights for further exploration into the mechanistic interactions of ORF3 and the host, as well as a possible role in pathogenesis in PEDV and other coronaviruses.

## 1. Introduction

Porcine epidemic diarrhea virus (PEDV) is the causative agent of a devastating swine enteric disease that presents with watery diarrhea, vomiting anorexia and severe dehydration among sows [1]. Since its early documentation in Europe, PEDV has spread to other parts of the globe with outbreak cases reported across many other countries in Southeast Asia as well as the Americas [2,3,4,5,6]. PEDV infects swine populations of all age groups with up to 100% mortality reported, especially among neonatal piglets, causing significant economic losses in the industry [7,8].

PEDV belongs to the coronaviridae family in the order Nidovirales [9]. Members of this group have a large, single-stranded RNA genome (approximately 28–30 kilobases) of positive polarity. The PEDV genome consists of five open reading frames (ORFs) downstream of ORF1, four of which code for structural proteins, including spike (S), membrane (M), small membrane (sM) and nucleocapsid (N) in order. The fifth is ORF3 located between S and M genes [10,11]. It is part of a group of unique small proteins encoded in coronaviruses termed as accessory proteins and believed to interfere with host immune response [12].

Adaptation of PEDV to culture results in mutations of the ORF3 gene. This has been associated with attenuation of the virus [13,14,15]. Additionally, mutations in ORF3 as well as spike genes resulting from increased passages contributed to reduced proliferation and pathogenesis of PEDV [16]. It has been demonstrated that the ORF3 gene of the prototype PEDV encodes an ion channel consisting of four transmembrane domains that are critical for virus production [17]. However, the laboratory strains are predicted to consist of transmembrane 1 and 2 between amino acids 40 and 98 but lack in transmembrane 3 and 4 [17,18]. This ORF3 protein aggregates in the cytoplasm, particularly in the endoplasmic reticulum (ER), inducing autophagy through endoplasmic stress [19]. Of note, ORF3-protein prolongs the S-phase of a mitotic cell [20].

Despite increasing evidence that implicates the ORF3 protein in virulence of PEDV, Kristen–Burmann et al. [21] suggest that ORF3 importance in pathogenicity is modest. To address such gaps in the knowledge of the ORF3 protein, there is a need to conduct continued investigations of the gene as well as the encoded protein. The current study reports critical cellular observations made in PEDV-ORF3_wt/N/C_-expressing cells. The analysis sheds light into the possible interactions of the ORF3 protein with the host cell.

## 2. Materials and Methods

### 2.1. Cell Culture

Mycoplasma-free cell-lines obtained from American Type Culture Collection (ATCC, Manassas, VA, USA) were employed in the study. These included African green monkey kidney cells, Vero (ATCC CCL-81), human embryonic kidney (HEK) 293T/17cells (ATCC CRL-11268), and mouse embryonic fibroblasts, NIH/3T3 (ATCC CRL-1658). Cell culture media and supplements were sourced from Thermo Fisher Scientific, Waltham, MA, USA. Cell culture protocols were described by ATCC with modifications. Briefly, minimum essential medium (MEM) supplemented with 5% of fetal bovine serum (FBS) and antibiotics (100 U penicillin and 100 µg streptomycin per milliliter, (Invitrogen) was used to culture Vero cells. For 293 T and NIH 3T3 cells, Dubbelco’s minimum essential medium with the above supplements was used. All cells were maintained at 37 °C, 5% CO_2_.

For plasmid cloning, *Escherichia coli* (*E. coli*) DH5 alpha cells were sourced (Invitrogen, CA, USA). Yeast extract and tryprone for bacteria media preparation were purchased from Intron Biotechnology.

### 2.2. Propagation of Cell-Adapted Pedv (Kpedv-9 Strain)

The cell-adapted strain of the Korean PEDV isolate, KPEDV-9, was maintained in the infectious diseases laboratory at Chungnam National University. For the purposes of obtaining the ORF3 gene, the virus was cultured in Vero cells as described, with some modifications [22]. Briefly, KPEDV-9 was inoculated into confluent Vero cells at a M.O.I. ≥ 1 and maintained at 37 °C, 5% CO_2_ for 1 h. Inoculum was removed and cultured in serum-free MEM at 37 °C, 5% CO_2_ for 48 h. The supernatant was harvested and stored at −70 °C. Progeny virions trapped in intracellular vesicles were released by repeated freeze-thawing. This was then centrifuged at 2000× *g* for 10 min at 4 °C and the cell debri free supernatant was collected. Virus concentration and partial purification were performed by ultracentrifugation under a 20% sucrose cushion at 28,000 rpm for 3.5 h. The pellet was re-suspended in 10 mM phosphate-buffered saline (PBS, pH 7.4) and stored at –70 °C.

### 2.3. ORF3 cDNA Synthesis

PEDV genomic RNA was extracted using the Ribo_spin vRD^TM^ kit following manufacturer’s instructions (GeneAll Biotechnology, Seoul, Republic of Korea). Reverse transcription ensued using SuperScript^TM^ III First-Strand Synthesis System as directed (Thermo Fisher Scientific, Waltham, MA, USA). Briefly, RNA was incubated together with Oligo (dT) primers and assorted deoxyribonuleotide triphosphates for 5 min at 65 °C to denature secondary structures. Next, reverse transcription of the first strand template was performed at 50 °C for 50 min before terminating the reaction at 85 °C for 5 min. Finally, reaction was chilled on ice before RNase treatment at 37 °C for 20 min.

### 2.4. Polymerase Chain Reaction (PCR)

ORF3, membrane and nuleocapsid genes were each amplified by PCR in a 50 μL mixture containing 100 ng of PEDV cDNA, gene specific primers of 1 μL (10 pM) each (Table 1), 2.5 μL (10 mM) of dNTPs, 5 μL of 10× PCR buffer (100 mM Tris–HCl (pH 8.8 at 25 °C), and 2.5 U of native *pfu* DNA polymerase enzyme (Solgent, Daejeon, Republic of Korea). The reaction was carried out using the following reaction cycles (Biometra TPersonal, Gottingen, Germany): initial denaturation at 95 °C for 5 min, followed by 35 cycles of denaturation, and primer annealing and fragment sunthesis at 72 °C. The amplification product was analyzed by electrophoresis on 1% of ethidium bromide-stained agarose gel.

### 2.5. Plasmid Constructs

ORF3, membrane, and nucleocapsid gene sequence were inserted into 5′ Bgl II/3′ Kpn I sites of pEGFP-C1 (Clonetech, CA, USA). The membrane and nucleocapsid proteins were used for comparative analysis of expression patterns against ORF3. Furthermore, the ORF3 gene was truncated following nucleotide analysis by the Simple Modular Architecture Research Tool (SMART) [23,24] as previously described [18]. Briefly, ORF3 was truncated into two fragments: N-term consisting of first 98 amino acids, (aa), and C-term, aa 99-224. In this model, two predicted transmembrane domains were in ORF3-N while ORF3-C had no transmembrane region. The sequences were designated as ORF3_wt_ (wild type), ORF3-N (N-terminal domain) and ORF3-C (C-terminal domain). These mutants were amplified by PCR using fragment specific primers (Table 1) and cloned into Bgl II and Kpn I sites of pEGFP-C1 for expression.

### 2.6. Sequencing

All PCR-amplified sequences were verified by sequencing (Solgent, Daejeon, Republic of Korea) using universal pEGFP-C1 primers (Table 2).

### 2.7. Expression of Orf3/Mutants, Membrane and Nucleocapsid

ORF3_wt_, ORF3-N and ORF3-C/pEGFP-C1 constructs were separately transfected in HEK-293T as well as NIH 3T3 cells. Expression of ORF3/mutant proteins was analyzed alongside GFP in an empty vector as a reference for transfection efficiency. Both HEK-293T and NIH 3T3 cell types were seeded in a six-well plate at a density of 2.5 × 10^5^/mL of the medium and incubated for 24 h at 37 °C and 5% CO_2_. The pEGFP-C1 empty vector (3 µg) as well as constructs containing ORF3_wt_, ORF3-N, ORF3-C, membrane or nucleocapsid were separately transfected into HEK-293T cells by the calcium phosphate method as described [25]. For NIH3T3 cells, ORF3_wt_/mutant constructs and pEGFP-C1 empty vector were independently lipo-transfected using Fugene HD (Roche, Basel, Switzerland), following the manufacturer’s instructions. Analysis of expression was performed in a time range of 6 h to 24 h post-transfection and visualized by GFP florescence at X40 microscopic magnification (Leica DM IL, Wetzlar, Germany).

### 2.8. Dapi Staining

Investigations of ORF3 impact on cellular DNA were performed by DAPI staining (Sigma-Aldrich, Saint Louis, MO, USA) as described [26]. Briefly, at 6, 12, 24, 36 and 42 h post-transfection, the cells were washed with 1× PBS and fixed with 5% of formaldehyde for 10 min. Cells were washed with 1× PBS and permeabilized with 0.1% Nonidet P-40 for 10 min. Cells were incubated with DAPI dissolved in 1× PBS at a final concentration of 1 µg/mL for 10 min in the dark. All above procedures were carried out at room temperature. Visualization was by florescence microscopy (Leica DM IL).

### 2.9. Cell Cytotoxicity Assay

Expression of ORF3_wt_ and mutants caused cell death. Cell lethality was evaluated by three replicate counts of dead cells floating in media using a haemocytometer (Abcam, Cambridge, England) without staining. The metabolic activities of ORF3_wt_/mutant-transfected cells were evaluated by MTT (3-[4,5-dimethythiazol-2-yl]-2,5-diphenyl tetrazolium bromide) reduction assay using the commercial Cyto X^TM^ kit (LPS Solution, Daejeon, Republic of Korea). HEK-293T cells were seeded (2.5 × 10^5^ cells/mL) in a six-well plate and transfected after 24 h with 3 µg of empty vector or pEGFP-C1 containing inserts of ORF3_wt_, ORF3-N or ORF3-C. At 24 h post-transfection, the cells were trypsin-treated and seeded at 10,000 per well density in a 96-well plate then incubated at 37 °C in 5% of CO_2_. Metabolic activity assays were carried out in a time range of 30 to 60 h. Briefly, cells were treated with 10 µL of cyto X solution for 4 h, after which optical density values were taken at 450 nm (Tecan Sunrise Remote, Mannedorf, Switzerland). As the positive control, cells were treated with 50 mM of hydrogen peroxide (H_2_O_2_), to induce oxidative stress, for 4 h. while the native cells (negative control) were treated with the transfection reagent alone.

## 3. Results

### 3.1. Orf3 Protein Localization Shifts from the Cytoplasm to the Nucleus

Florescence microscopy of HEK-293T cells expressing GFP alone or in fusion with ORF3_wt_ or its truncated mutants showed that the ORF3_wt_/ORF3-N/ORF3-C proteins had initial expression in the cytosol (6 h post-transfection). However, ORF3_wt_ and ORF3-N began to aggregate around the nucleus as observed from 12 h of transfection and continued afterwards. Similarly, ORF3-C demonstrated nuclear localization but to a lesser extent. On the other hand, GFP protein expressed and remained in the cytoplasm (Figure 1).

### 3.2. Orf3 Protein Causes Nuclear Condensation and Morphological Distortion

Analyses of expression of the various ORF3 proteins by U.V. microscopy revealed that ORF3 reduced the expression efficiency of GFP. Furthermore, ORF3/mutant-expressing cells had distorted morphology (Figure 2A). Analyses of DAPI-stained HEK-293T and NIH 3T3 cells revealed ORF3_wt_ and that its truncated mutants condensed the nucleus following accumulation of these proteins in the nucleus. Merging of U.V. and DAPI images suggested that ORF3 proteins had sequestered the nuclei of host cells. Furthermore, the nuclear condensation was associated with cell rounding and shrinking. From morphological observation of the cells, ORF3-N exhibited more cellular damage than ORF3_wt_ and ORF3-C in order, especially apparent in the HEK-293T cells. While GFP alone also appeared to be present in the nucleus, no adverse effect was observed on the nuclei or morphology of the cells (Figure 2 and Figure 3).

### 3.3. Orf3 Proteins Induced Cell Mortality in Expressed Cells

After 24 h post-transfection of the HEK-293T cells, fluorescence microscopy revealed that some ORF_wt_, ORF3-N and ORF3-C-expressing cells were detached from substratum, indicating death. The floating cells were still fluorescing green. In contrast, cell death did not occur with the expression of GFP alone. Similarly to observations on cell damage, ORF3-N was associated with the most cell lethality followed by ORF_wt_ and ORF3-C in order. The mean number of dead cells was determined at about 2.46 × 10^4^ for HEK-293T/ORF_wt_, 7.38 × 10^4^ for HEK-293T/ORF3-N, and 2.3 × 10^4^ for HEK-293T cells alone as well as HEK-293T/GFP and HEK-293T/ORF3-C cells. Relative to the native cells, ORF3_wt_ increased cell death by 6.61%, ORF3-N caused 220.32% increased lethality, and ORF3-C had 0.64% more dead cells (Figure 4).

### 3.4. Orf3 Down-Regulated Cell Metabolism

Analyses of mitochondrial succinate dehydrogenase by cell treatment with a reducible chromogenic substrate, MTT, showed that the O.D_450_ values were relatively lower in GFP-fused ORF3_wt_, ORF3-N or ORF3-C-expressing cells than those expressing GFP only, or native cells. In reference to the native cells, at 30 h of expression, ORF3_wt_ had 15% decreased metabolic activity, ORF3-N decreased by 11%, and ORF3-C was lower by 14%. At 48 h, cellular metabolism was determined to be 54% for ORF3_wt_-expressing cells, 56% for ORF3-N, and 68% for ORF3-C. At 60 h, the metabolic activities recorded were 59%, 64% and 81% for ORF3_wt,_ ORF3-N and ORF3-C-expressing cells, respectively. (Figure 5).

## 4. Discussion

The product encoded by the ORF3 gene of PEDV has been speculated to be important for pathogenicity [12]. We have previously reported that the ORF3 protein suffers host inhibition [18], as independently observed [27]. On the contrary, PEDV structural genes of membrane and nucleocapsid expressed the respective proteins readily. HEK-293T cells expressing membrane, nucleocapsid, ORF3 or its truncation mutants revealed cytoplasmic localization of the PEDV proteins. This was consistent with other independent investigators [19]. However, unlike membrane and nucleocapsid, ORF3/mutant proteins’ cytosolic expression was transitory with observations of aggregation in the nucleus as from 12 h onwards.

Microscopic examination of DAPI-stained DNA demonstrated nuclear condensation and disintegration. Though merged images of the green florescent and blue DAPI-stained images did not show complete overlap of the two, the blue and green co-appeared within the nuclear space. This led to a postulation that the ORF3 protein may actively interact with nuclear membrane. Some members of rhabdoviruses can exemplify this imagination albeit in reverse manner. These nucleorhabdoviruses display a nucleocapsid-coated granular matrix, which buds at the inner nuclear membrane, and accumulate in the perinuclear space [28]. It is known that the PEDV-ORF3 protein traffics to and accumulates in the Golgi area via the exocytic pathway. Moreover, the C-terminal consists of an essential motif for transport from the ER without which the ORF3 protein is retained in the ER as well as ER-to-Golgi intermediate compartment [29]. This could explain our observations of ORF3 translocation from the cytoplasm and accumulation around the nucleus. Furthermore, proteomic analyses have demonstrated that the PEDV-ORF3 protein interacts with host proteins mainly in the endo-lysosomal organelles with consequent interference of immune signaling [30,31].

On the other hand, GFP also appeared in the nucleus. Indeed, GFP has been shown to escape nuclear membrane blockage due to its small size (27 kDa). The nuclear pore complex facilitates entry of small molecules into the nucleus and GFP has been found to diffuse bi-directionally via this portal [32]. However, GFP was shown to neither affect the function of co-expressed proteins nor cell viability by Marshall et al. [33]. Other independent investigators documented the suitability of GFP fusion proteins in the study of subcellular localization as well as transient expression assays, revealing that the larger molecules could not penetrate the nuclear envelop by simple diffusion. In particular, the GFP-DETI protein had limited nuclear localization, demonstrating that GFP cannot transport a fusion protein to the nucleus. The nuclear localization of GFP-fused proteins is only possible by active or facilitated transport. As such, GFP-COP1 localized to discrete subnuclear particles and the GFP did not affect the function of COP1 consistent with Marshall et al. [34].

Although PEDV-ORF3 protein has been reported to have a limited role in pathogenesis [21], our findings suggest the contrary. ORF3_wt_ and ORF3-N, in particular, caused nuclear condensation, morphological distortion and cell death. Though a cell count may not discriminate a viable floating cell from a dead one, the marked difference in the count in ORF3_wt_ and ORF3-N should be of curious interest. Similarly, ORF3 protein was reported to cause endoplasmic stress that resulted in the disruption of ER and cell membrane, leading to cell death [19]. While nuclear condensation is likely to occur from apoptosis, Zou et al. [19] demonstrated that the ORF3-protein does not trigger apoptosis. Thus, ORF3-protein perceivably causes necrotic cell death, which could be critical for causing intestinal lesions.

To investigate the probable cause of cell death, metabolic activity was assayed. ORF3_wt_ and ORF3-N in particular down-regulated host metabolism, and caused nuclear condensation, morphological distortion and cell death. Similarly, ORF3 protein was independently reported to cause endoplasmic stress that resulted in the disruption of ER and cell membrane, leading to cell death [19]. While nuclear condensation is likely to occur from apoptosis, Zou et al. [19] demonstrated that ORF3-protein does not trigger apoptosis. Thus, ORF3-protein perceivably causes necrotic cell death, which could be critical for causing intestinal lesions. It is not clear how ORF3-protein down-regulated metabolism but with endoplasmic stress, cell physiology could be sub-optimized.

## 5. Conclusions

It is noteworthy that while our findings attribute importance to the N-terminal domain, and the circulating ORF3 gene is shown to have four transmembrane domains [17]. These together form an important ion channel. Transmembrane 3 and 4 occur in the C-terminal of our model, and are lost upon culture adaptation, reducing ion channel activity and the pathogenicity of the virus. Our results attribute the ORF3 impact to the transmembrane domains in ORF3-N. The impact of ORF3_wt_ which invariably has the transmembrane was not as high. This can be explained by inhibited expression of the whole gene (18) rather than diverged pattern of function. Taken together, we conclude that the function of ORF3-protein is inherent in its transmembrane domains and that all the four transmembrane domains are critical for activity and virulence.

## Figures and Tables

**Figure 1 viruses-17-01468-f001:**
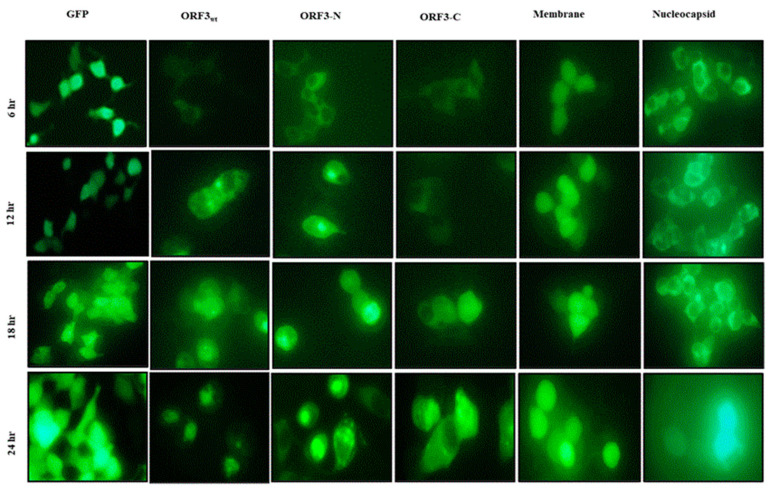
**U.V. microscopy of HEK-293T cells.** HEK-293T cells expressing GFP protein alone or GFP fused with ORF3_wt_/ ORF3-N/ORF3-C/ membrane/nucleocapsid proteins. GFP protein alone was examined at ×200 magnification due to high expression efficiency and glow. The PEDV proteins were analyzed at ×400 magnification. Early (6 h) cytoplasmic localization of ORF3/ ORF3-N proteins was observed in the cytoplasm but at 12 h, these proteins exhibited accumulation around the nucleus, proceeding to aggregate in the nucleus at 24 h. ORF3-C had relatively lower nuclear accumulation relative to ORF3_wt_ and ORF3-N. The GFP, membrane and nucleocapsid proteins were observed in the cytosol with no shift to the nucleus.

**Figure 2 viruses-17-01468-f002:**
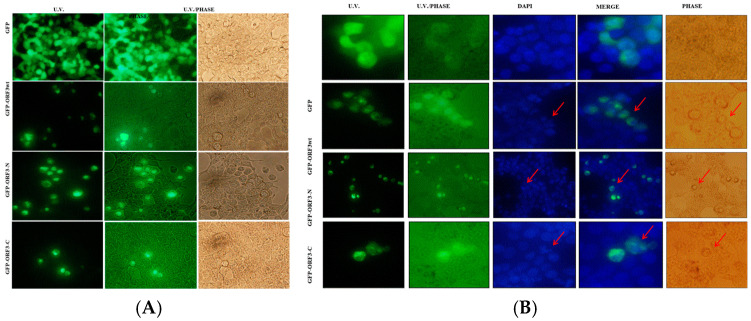
**ORF3 expression causes morphological and nucleus distortion.** Microscope images were taken at ×200 magnification. In (**A**), about 90% of HEK-293T live cells expressing GFP alone had a normal morphology. Transfection with ORF3_wt_/ORF3-N/ORF3-C-fused GFP proteins had reduced transfection efficiency. The expression of ORF3 or truncated mutants resulted in shape distortion (rounding). ORF3-C effect on cells was low compared to the ORF3 and wild type. In (**B**), fixed DAPI-stained HEK-293T cells expressing GFP protein alone or GFP fused with ORF3_wt_/ ORF3-N/ORF3-C proteins were analyzed. U.V. microscopy revealed GFP florescence; ORF3_wt_/ORF3-N/ORF3-C-fused GFP proteins exhibited nuclear localization. DAPI stain revealed nuclei condensation in GFP-ORF3/mutant-expressing cells but not in cells expressing GFP alone. The merged images show that ORF3/mutant proteins sequester the nucleic material. The U.V./phase and phase images together reveal that the GFP-fluorescing cells are morphologically distorted (rounding) where fused with ORF3_wt_/ORF3-N/ORF3-C but not in GFP alone. However, ORF3-C causes lesser distortion in comparison with ORF3_wt_ and ORF3-N. Arrows point to the condensed nuclei (DAPI) or rounded cells (phase). GFP control also appears to be present in the nucleus but the DAPI-stained nucleus remained well rounded and no morphology anomalies were observed in phase images.

**Figure 3 viruses-17-01468-f003:**
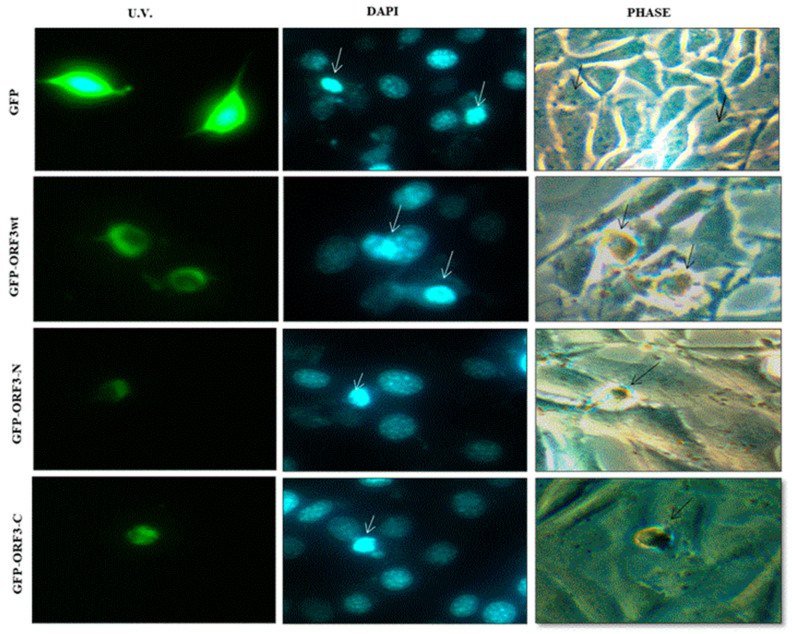
**DAPI stained NIH 3T3.** Fixed DAPI stained NIH 3T3 cells expressing GFP protein alone or GFP fused with ORF3_wt_/ ORF3-N/ORF3-C proteins. U.V. microscopy reveals GFP florescence in the cytoplasm; GFP-ORF3_wt_/ORF3-N/ORF3-C exhibits nuclear localization. DAPI stain reveals nucleic appearance in GFP-ORF3/mutant-expressing cells comparable to GFP-expressing cells (shown by white arrows). ORF3_wt_ / ORF3-N and ORF3-C expression has associated morphological distortion evidenced in the phase images (shown by black arrows). NIH 3T3 cells expressing GFP alone are not morphologically distorted.

**Figure 4 viruses-17-01468-f004:**
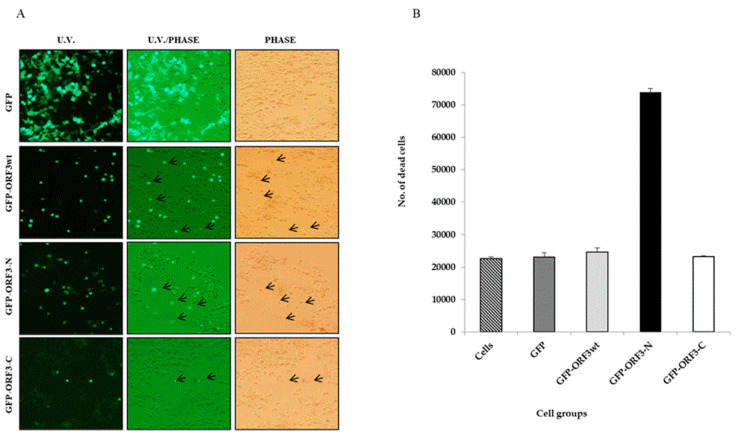
**ORF3 expression has associated cell lethality.** (**A**). Observation of GFP- and GFP-tagged ORF3_wt_ or mutants under U.V. microscope. U.V. and light together (U.V./light) and under light microscope (phase). Green flourescing cells were observed floating in media of ORF3_wt_ and ORF3-N-expressing HEK-293T cells. Arrows point to floating cells which were considered as dead. (**B**). Value of means of counted dead cells from each cell group. Cell counting was performed twice per cell group using a hemocytometer and reported as mean ± SD (standard deviation). Results of ORF3-C-expressing cells were comparable to GFP alone but the detached cells were increased where ORF3_wt_ or ORF3-N was expressed with ORF3-N causing the greatest impact to cells.

**Figure 5 viruses-17-01468-f005:**
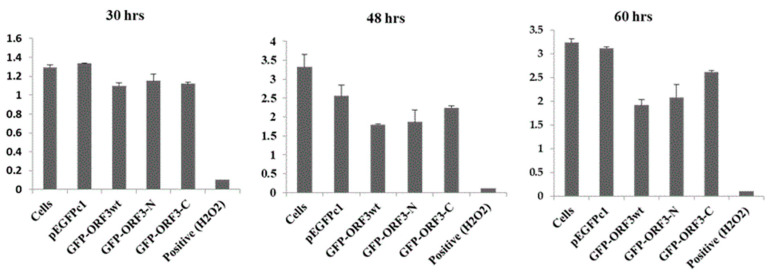
**Quantitative activities of mitochondrial succinate dehydroagenase.** MTT assay of 293T cells treated with 50 mM of H_2_O_2_ for 4 h (positive); GFP, GFP/ORF3_wt_, ORF3-N or ORF3-C transfections and O.D._450_ values determined at the indicated time points. Cells were treated with transfection reagents without DNA for negative control. Expression of OFR3 proteins negatively impacted on host cell metabolism; ORF3_wt_ had the greatest impact (44% average reduction) followed by ORF3-N (average 30%) and ORF3-C (average 22%). The positive control was for the validation of the MTT assay. The values are reported as means of three replicates ± SD.

**Table 1 viruses-17-01468-t001:** Primer sequences used for amplification of various PEDV genes and fragments.

	Primer	Orientation	Sequence
ORF3_wt_ ORF3-N ORF3-C Membrane Nucleocapsid	Accession no. NC_003436 MK458321 KF898123	Forward Reverse Forward Reverse Forward Reverse Forward Reverse Forward Reverse	5′-ATG TTT CTT GGA CTT TTT 5′-GCT TCA ATT AGT GAA TGA 5′-ATG TTT CTT GGA CTT TTT 5′-AATAAT AGT TGC ATC TAA 5′-ATG TGT TGC ACA CTT ATT GGC 5′-GCT TCA ATT AGT GAA TGA 5′-ATGTCT AACG GTTCT ATTCC CG 5′-TTAGACTAAATGAA GCAC 5′-ATGGCTT CTG TCA GCTTTC 5′-TTA ATT TCC TGT ATC GAAG

**Table 2 viruses-17-01468-t002:** Primer sequences used for sequencing of ORF3_wt_ and ORF3-C/ORF3-N.

	Primer	Orientation	Sequence
pEGFP-C1	EGFP-CF SV40-pAR	Forward Reverse	5′-TAA TAC GAC TCA CTA TAGG

## Data Availability

Data will be provided on request.

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
