# Peer review of "ORF3 Gene of Porcine Epidemic Diarrhea Virus Causes Nuclear and Morphological Distortions with Associated Cell Death"

_viruses, 2025, doi:10.3390/v17111468_

Round 1
Reviewer 1 Report
Comments and Suggestions for Authors
I am honored to read this article. Based on the data shown in the manuscript, I do believe that this study can be helpful to illustrate the function of PEDV ORF3 protein. However,due to the small volume and poor quality of the data, I do not think this article meet all the requirements of MDPI at present. Therefore, I would like to recommend a major revision before the potential acceptance for publication in viruses. The following are some suggestions regarding this article.
- For figure 1, in order to better localize the ORF3 protein, it is necessary to stain the nucleus. Since the authors used GFP as a fluorescent marker, a live-cell nuclear stain (such as hoechst 33342) can be selected for nuclear staining.
- According to Figure 2, some GFP also appears to localize in the nucleus. The authors need to rule out the possibility that the nuclear localization of ORF3 is caused by GFP. In addition, the expression levels of the four proteins vary significantly. The authors need to design experiments to verify whether this is the cause of the differences in the subsequent results.
- All images in the article are very blurry and must be replaced with clearer ones if it is to be published. In addition, the magnification scales are inconsistent across different figures. Moreover, it is necessary to indicate the magnification in the figure captions and add scale bars in each picture.
- The data presented in this article are all in the form of images, lacking quantitative analysis. The data volume in this article is too small and overly simplistic. It is recommended that the author continue to investigate the mechanism by which ORF3 nuclear localization causes cell death in order to enrich the content of this paper.
- For figure 4,it is recommended to use flow cytometry to detect the number of the detached cells.
- For figure 2, the labels on the left are misplaced.
- Line 66: There is a problem with the paragraph indentation.
- Line 108 “2.4 Polymerase Chain Reaction (Pcr)” →“2.4 Polymerase Chain Reaction (PCR)”.
- Line 110: “1 μL (10 μM) each (Table 1)”?
- Table 2 appears to be missing one primer.
- Line 262: The citation format is incorrect (18).
Author Response
Reviewer 1
I am honored to read this article. Based on the data shown in the manuscript, I do believe that this study can be helpful to illustrate the function of PEDV ORF3 protein. However,due to the small volume and poor quality of the data, I do not think this article meet all the requirements of MDPI at present. Therefore, I would like to recommend a major revision before the potential acceptance for publication in viruses. The following are some suggestions regarding this article.
- For figure 1, in order to better localize the ORF3 protein, it is necessary to stain the nucleus. Since the authors used GFP as a fluorescent marker, a live-cell nuclear stain (such as hoechst 33342) can be selected for nuclear staining.
We are grateful for taking time to review our manuscript and for your valuable suggestion.
Response.
If you allow, the early observations reveal a clear demarcation of the nucleus as a dark core that did not fluoresce.
Preliminary expression in NIH 3T3 cells showed similar results (x400 magnification)
|
- According to Figure 2, some GFP also appears to localize in the nucleus. The authors need to rule out the possibility that the nuclear localization of ORF3 is caused by GFP. In addition, the expression levels of the four proteins vary significantly. The authors need to design experiments to verify whether this is the cause of the differences in the subsequent results.
Response
Much appreciated. GFP was shown to neither affect the function of co-expressed proteins nor cell viability by Marshall et al., 1995. Our results show that GFP also localized in the nucleus and your concern is valid. Indeed, GFP has been shown to escape nuclear membrane blockage due to its small size (27 kDa). The nuclear pore complex (NPC) facilitates entry of small molecules into the nucleus and GFP has been found to diffuse bi-directionally via this portal (Wei et al., 2003).
However, this concerns have been addressed independently and the suitability of GFP fusion proteins in the study of subcellular localization as well as transient expression assays revealed that the larger molecules could not penetrate the nuclear envelop by simple diffusion. In particular, the GFP-DETI protein had limited nuclear localization demonstrating that GFP cannot transport a fusion protein to the nucleus. The nuclear localization of GFP fused proteins is only possible by active or facilitated transport. In contrast, GFP-COP1 localized to discrete sub-nuclear particles and the GFP did not affect the function of COP1 consistent with Marshall et al. (Von Arnim et al., 1998)
On expression levels, it is true that the levels were different. Initial experiments did not yield any expression results of the wild type and the truncation was arrived at as a way of circumventing the resistance as earlier reported (Kamau et al., 2024).
If agreeable, we have discussed our findings as such.
References
-Marshall J, Molloy R, Moss GW, Howe JR, Hughes TE. The jellyfish green fluorescent protein: a new tool for studying ion channel expression and function. Neuron. 1995 Feb;14(2):211-5. doi: 10.1016/0896-6273(95)90279-1.
-Wei, X., Henke, V. G., Strübing, C., Brown, E. B., & Clapham, D. E. (2003). Real-time imaging of nuclear permeation by EGFP in single intact cells. Biophysical journal, 84(2), 1317-1327. https://doi.org/10.1016/S0006-3495(03)74947-9
-Von Arnim, A. G., Deng, X. W., & Stacey, M. G. (1998). Cloning vectors for the expression of green fluorescent protein fusion proteins in transgenic plants. Gene, 221(1), 35-43.
- Kamau, A. N., Yu, J. E., Park, E. S., Rho, J. R., Hong, E. J., & Shin, H. J. (2024). Strenuous expression of porcine epidemic diarrhea virus ORF3 protein suggests host resistance. Veterinary Microbiology, 297, 110193.
- All images in the article are very blurry and must be replaced with clearer ones if it is to be published. In addition, the magnification scales are inconsistent across different figures. Moreover, it is necessary to indicate the magnification in the figure captions and add scale bars in each picture.
Response
Images have been revised and magnification scales indicated.
- The data presented in this article are all in the form of images, lacking quantitative analysis. The data volume in this article is too small and overly simplistic. It is recommended that the author continue to investigate the mechanism by which ORF3 nuclear localization causes cell death in order to enrich the content of this paper.
Response
Other data sets have been included to add weight to the content.
- For figure 4,it is recommended to use flow cytometry to detect the number of the detached cells.
Response
Unfortunately, we are not able to analyze as suggested hope we can be excused on this.
- For figure 2, the labels on the left are misplaced.
Response
The figure has been done a fresh and that has been rectified.
- Line 66: There is a problem with the paragraph indentation.
Response
The line has changed but I believe it is the paragraph starting with “Despite increasing evidence,…”.
That has been corrected.
- Line 108 “2.4 Polymerase Chain Reaction (Pcr)” →“2.4 Polymerase Chain Reaction (PCR)”.
Response
Corrected
- Line 110: “1 μL (10 μM) each (Table 1)”?
Response; Corrected to10 pM
- Table 2 appears to be missing one primer.
Response; sequence added
- Line 262: The citation format is incorrect (18).
Response; Corrections made.

Reviewer 2 Report
Comments and Suggestions for Authors
This study investigated the functions of the ORF3 gene of the Porcine epidemic diarrhoea virus (PEDV), a swine disease that can cause high mortality among young piglets. The study concluded that the ORF3 gene encodes a protein that causes nuclear damage, distorts cell morphology and leads to cell death. These findings underpin the importance of the ORF3 gene expression in the host and are rudimentary insights for further exploration into the mechanistic interactions of ORF3 and the host, as well as the possible role in pathogenesis in
PEDV and other coronaviruses. This study provides useful information on the role of the ORF3 gene of the PEDV and would assist future studies investigating the pathogenesis and virulence of PEDV. The study design is appropriate, and the conclusions support the study hypothesis. After a careful review, I have only a few minor comments:
Line 131-132: Please provide more information on the sequencing strategy and technology used.
Line 144: Was the Western blot performed to analyse the expression?
Author Response
Reviewer 2
his study investigated the functions of the ORF3 gene of the Porcine epidemic diarrhoea virus (PEDV), a swine disease that can cause high mortality among young piglets. The study concluded that the ORF3 gene encodes a protein that causes nuclear damage, distorts cell morphology and leads to cell death. These findings underpin the importance of the ORF3 gene expression in the host and are rudimentary insights for further exploration into the mechanistic interactions of ORF3 and the host, as well as the possible role in pathogenesis in
PEDV and other coronaviruses. This study provides useful information on the role of the ORF3 gene of the PEDV and would assist future studies investigating the pathogenesis and virulence of PEDV. The study design is appropriate, and the conclusions support the study hypothesis. After a careful review, I have only a few minor comments:
Line 131-132: Please provide more information on the sequencing strategy and technology used.
Response. The sequencing method has been stated.
Line 144: Was the Western blot performed to analyse the expression?
The expression of ORF3 was faced with difficulty and the eventual success results including western blot data were previously published.
(Kamau, A. N., Yu, J. E., Park, E. S., Rho, J. R., Hong, E. J., & Shin, H. J. (2024). Strenuous expression of porcine epidemic diarrhea virus ORF3 protein suggests host resistance. Veterinary Microbiology, 297, 110193).
